# SMASH: Scalable Method for Analyzing Spatial Heterogeneity of genes in spatial transcriptomics data

**Souvik Seal** [1]*, **Benjamin G. Bitler**[2], **Debashis Ghosh**[3]

**1** Department of Public Health Sciences, School of Medicine, Medical University of South Carolina, Charleston, South Carolina, United States of America, **2** Department of Obstetrics and Gynecology, School of Medicine, University of Colorado Denver Anschutz Medical Campus, Aurora, Colorado, United States of America, **3** Department of Biostatistics and Informatics, Colorado School of Public Health, University of Colorado Denver Anschutz Medical Campus, Aurora, Colorado, United States of America

* sealso@musc.edu

**Data Availability Statement:** A Python-based software implementation of SMASH is available at, https://github.com/sealx017/SMASH-package. The package provides two detailed notebooks to perform the analysis on the mouse hypothalamus

## Abstract

In high-throughput spatial transcriptomics (ST) studies, it is of great interest to identify the genes whose level of expression in a tissue covaries with the spatial location of cells/spots. Such genes, also known as spatially variable genes (SVGs), can be crucial to the biological understanding of both structural and functional characteristics of complex tissues. Existing methods for detecting SVGs either suffer from huge computational demand or significantly lack statistical power. We propose a non-parametric method termed SMASH that achieves a balance between the above two problems. We compare SMASH with other existing methods in varying simulation scenarios demonstrating its superior statistical power and robustness. We apply the method to four ST datasets from different platforms uncovering interesting biological insights.

## Author summary

In recent years, spatial transcriptomics (ST) has become increasingly popular to study the expression profile of genes across different spatial locations of a tissue. Many of the genes exhibit spatially varying expression patterns making them immensely valuable for understanding the structural and functional properties of the tissue. The proposed method termed SMASH enables powerful and scalable detection of such genes in high-dimensional ST datasets.

## Introduction

Spatial transcriptomics (ST) performs high-throughput measurement of transcriptomes in complex biological tissues at single-cell or subcellular resolution, preserving spatial information [1–9]. In the past decade, the rapid development of ST technologies has facilitated exciting discoveries in different domains, including neuroscience [10–12] and cancer research [13–15].

data by MERFISH and the human DLPFC data by 10X Visium (along with the datasets as compressed Python objects). Both the mouse cerebellum data by Slide-seqV2 and the human DLPFC data by 10X Visium are available in the R Bioconductor package: STexampleData, available at, https://bioconductor.org/packages/release/data/experiment/html/STexampleData.html. The full mouse hypothalamus data by MERFISH is available at the link provided in the corresponding manuscript, from which we focused on only "Replicate 6", as it had the largest number of cells. The SCCOHT dataset by 10X Visium was collected at the University of Colorado Denver Anschutz Medical Campus, and is provided in the Github repository.

**Funding:** S.S. was supported in part by the Biostatistics Shared Resource, Hollings Cancer Center, Medical University of South Carolina (P30 CA138313). The funders had no role in study design, data collection and analysis, decision to publish, or preparation of the manuscript.

**Competing interests:** The authors have declared that no competing interests exist.

The popular ST technologies and corresponding platforms differ in terms of the procedure used to record spatial profiles, such as region of interest (ROI) selection [16, 17], next-generation sequencing (NGS) with spatial barcoding [18–20], and single-molecule fluorescence in situ hybridization (smFISH) [21–23]. Two crucial aspects that a researcher considers before choosing a suitable platform, are a) the capability of transcriptome-wide profiling, and b) the granularity of spatial resolution. For example, the majority of the smFISH-based technologies excel at capturing single-cell level resolution but lack the capability of transcriptome-wide profiling. On the other hand, ROI or NGS-based technologies can be used for transcriptome-wide profiling but on a significantly lower spatial resolution, such as 55 $\mu$m for the most popular and commercialized ST platform Visium (10X Genomics). We refer to Moses et al. (2022) [24] for a detailed discussion on these technologies. Deriving biological insights from datasets obtained using such platforms with either huge spatial or genomic profiles, or both, not only poses numerous statistical challenges but also requires maximum computational efficiency [25].

A critical step in the analysis of ST datasets is to identify the genes whose level of expression co-varies with the spatial locations across the tissue. These genes, often referred to as spatially variable genes (SVGs), can be used in downstream analyses, such as identifying potential markers for biological processes and defining areas in the tissue that dictate cellular differentiation and function [26–29]. For example, Wang et al. (2020) [30] analyzed an ST dataset on the tumor microenvironment (TME) of three tissue sections from a prostate cancer subject [31]. In every tissue section, a unique set of spatially variable metabolic genes were identified, which could arguably be used to guide targeted tissue-specific therapy. A simplistic approach for detecting SVGs could be to identify spatially located layers or cell types (if any) based on either a priori biological knowledge or using popular software, such as RCTD [32] and Seurat [33], with the transcriptional profiles, and then checking which genes exhibit highly enriched expression in a particular spatial layer or cell type. However, such an approach would achieve satisfactory performance only if the layers or cell types are spatially well-separated, and always be sensitive to the quality of the layer or cell type-identification step [34]. In recent years, more sophisticated methods have been developed to identify SVGs, a systematic overview of some of which can be found in Li et al. (2021) [35]. The methods can be broadly classified into three types: a) based on statistical modeling, b) based on machine learning or neural network, and c) based on graphical networks or spatial grids. Some of the notable methods of each type are, type (a): Trendsceek [36], SpatialDE [34], SPARK [37], SPARK-X [38], Boost-GP [39], and nnSVG [40], type (b): SPADE [41], SOMDE [33], and SpaGCN [42], and type (c): HMRF [43], MERINGUE [44], Binspect-Giotto [45], Boost-MI [46], ScGCO [47], and SpaGene [48]. We focus on methods of types (a) and (c) in this manuscript.

The statistical power of the methods greatly varies based on gene expression patterns and the spatial structure of ST datasets. The methods encounter different levels of computational complexity based on two quantities, $N$ and $K$, denoting the numbers of cells/spots and genes, respectively. SpatialDE [34] is one of the earliest methods of type (a). It employs a Gaussian process (GP) regression model [49] with kernel-based covariance matrices [50] of multiple types, such as linear, Gaussian, and cosine, computed using the distance between the spatial coordinates of the cells. The model decomposes the total variability of a gene expression into two components, spatial and error variance. A significantly large value of the spatial variance would imply that the gene is spatially variable. Borrowing an efficient estimation algorithm from the statistical genetics literature [51], SpatialDE manages to estimate the variance components with a reasonable degree of computational efficiency, requiring $O(N^3 + N^2K)$ floating point operations (FLOPS). A newer method named SPARK [37] extends the framework of SpatialDE by considering a generalized linear spatial model (GLSM) [52] with a Poisson distribution, arguing to be better suited for modeling the raw count data from the ST platforms

directly. However, the penalized quasi-likelihood (PQL) approach [53] used for parameter estimation in SPARK is extremely computationally demanding with a complexity of $O(N^3K)$, making it unusable for a transcriptome-wide analysis when $N$ is moderately large ($N > 3,000$). To this end, a non-parametric highly scalable method named SPARK-X [38] has been recently developed requiring just linear complexity w.r.t. $N$. It is based on the robust covariance testing framework [54] that compares the linear kernel-based covariance matrices of the gene expression and the spatial coordinates. However, using a linear kernel makes SPARK-X equivalent to fitting a multiple linear regression model [55] with the gene expression as the dependent variable and the spatial coordinates (or, some transformation of these) as the predictors and testing if the fixed effect coefficients differ from zero. Thus, it is only capable of detecting spatial dependencies or patterns that manifest linearly in the mean or expected value of the gene expression, also known as first-order dependencies, and drastically loses power in complex scenarios as to be shown later. Zhu et al. (2021) [38] has partially acknowledged this issue with their primary focus being computational scalability.

On the other side, a popular method of type (c), MERINGUE [44] considers spatial auto-correlation and cross-correlation based on spatial neighborhood graphs to identify SVGs. Improving hugely on the complexity of MERINGUE, another model-free method named Spa-Gene [48] has been recently developed. It constructs a spatial network between cells/spots using the $k$-nearest neighbors approach, and then for each gene, extracts the subnetwork whose nodes have high gene expression. Then, it compares the observed degree distribution of the subnetwork to a distribution from a fully connected network using the earth mover's distance [56]. It considers a permutation test [57] to obtain the $p$-value for every gene. SpaGene is highly comparable to SPARK-X w.r.t. computational complexity and thus applicable to ST datasets with large $N$. However, the method is harder to interpret than the methods of type (a), can not readily accommodate additional covariates, and also lacks power in various scenarios (see Simulations section).

We propose a non-parametric method, named SMASH, which achieves superior statistical power than both SPARK-X and SpaGene, while remaining computationally tractable. It augments the idea of SPARK-X in its use of the Hilbert-Schmidt independence criteria (HSIC) or robust covariance testing framework [54, 58] coupled with more general kernel-based spatial covariance matrices. With a computational complexity quadratic in $N$, SMASH sacrifices some degree of computational efficiency in favor of significantly higher detection power than both SPARK-X and SpaGene. However, it is worth highlighting that SMASH is notably faster than other type (a) methods, such as SpatialDE and SPARK, and can thus be thought of as a balanced alternative, fusing high detection power with a moderate degree of scalability. In varying simulation scenarios, we demonstrate that SMASH achieves highly consistent and superior performance as compared to the methods SPARK-X and SpaGene. Finally, our analysis of four large ST datasets from platforms like SlideSeq V2, Visium, and MERFISH using these three methods, not only reveals exciting biological insights but also demonstrates SMASH's capability of detecting SVGs that will be otherwise missed by either of the other two methods. A Python-based software implementation of SMASH is available at, https://github.com/sealx017/SMASH-package, which returns the lists of SVGs detected by both SMASH and SPARK-X, allowing users to investigate the overlap between them.

## Results

### Simulations

We evaluated the performance of SMASH, SPARK, and SpaGene in three different simulation studies. We omitted SpatialDE and SPARK from the power comparison for two reasons: a)

high computational requirements and b) these two methods have already been thoroughly studied in previous works [38, 48]. In simulation setup (1), we followed the procedure described in the SPARK-X manuscript [38]. In setups (2) and (3), we considered the Gaussian process (GP)-based spatial regression model from the SpatialDE manuscript [34], respectively with the Gaussian and cosine kernel-based covariance functions (see Eq (1)). In all the setups, three values of the number of cells ($N$) were considered, $N = 1000$, 5000, and 10,000. The spatial coordinates of the cells were simulated first, followed by the expression levels of $K$ (500 or 1000) genes with varying levels of dependence. In setup (1), the expression levels were simulated using a negative binomial distribution, while in setups (2) and (3), the expression levels were simulated using a multivariate normal distribution. In all the setups, distinct spatial patterns were ensured to be present in the expression levels. Further details regarding the simulation setups are provided at the end of the Methods section. Figs 1, 2 and 3 respectively correspond to the three simulation setups, in which we display the simulated spatial patterns and the statistical power of the three methods for different parameter combinations.

In simulation setup (1), SMASH, and SPARK-X performed much better than SpaGene for all four spatial patterns, namely streak, reverse streak, hotspot, and reverse hotspot (Fig 1). SpaGene was particularly poor for the patterns: streak and hotspot. The power of SMASH and SPARK-X steadily increased as $N$ and the fold-change parameter increased. Note that a fold value of 1 implied no spatial association while a larger value indicated higher spatial association. This particular simulation setup favored SPARK-X in the sense that the spatial variability of the expression was of the first order, manifesting entirely through the mean or expectation. Even in this scenario, SMASH managed to achieve similar power.

In simulation setups (2) and (3), the spatial variability of the expression was of higher order, manifesting through the covariance. In setup (2), which involved the Gaussian covariance function, SMASH performed the best followed by SPARK-X and then SpaGene in most cases. SMASH performed the best in setup (3) as well. However, SpaGene achieved better power than SPARK-X here. SPARK-X had almost zero power in many of the cases, especially when the period $p$ was small ($p = 0.5, 1$), demonstrating its lack of robustness under complicated spatial dependency structures.

We compared the run-time of the methods in the simulation setup (2) for varying numbers of cells, $N = 1000$, 5000, and 10000 (Table 1). Since the computational complexity of the algorithms mainly differs w.r.t. $N$ and not the number of genes $K$, we kept $K = 1000$. We noticed that the run-time of SMASH expectedly increased in an almost squared order w.r.t. $N$. SPARK-X and SpaGene were both extremely fast for just having linear complexity w.r.t. $N$. We also added SpatialDE to this comparison to show how computationally intensive it can be to fit a fully parametric model in such a context. We omitted SPARK entirely as it is much slower than even SpatialDE with a computational complexity of $O(N^3K)$.

## Application to real data

We applied the methods, SMASH, SPARK-X, and SpaGene to four datasets: 1) mouse cerebellum data collected using Slide-seq V2 [19, 59], 2) human dorsolateral prefrontal cortex (DLPFC) data collected using Visium [11], 3) small cell ovarian carcinoma of the ovary hypercalcemic type (SCCOHT) data collected using Visium [11], and 4) mouse hypothalamus data collected using MERFISH [60, 61]. The datasets have varying numbers of genes and spots/cells.

**Mouse cerebellum by Slide-seqV2.** The mouse cerebellum data [19] has 20,117 genes and 11,626 spots. We restricted our focus to the 7,653 genes that express in more than 1% of the spots. The mouse cerebellum is made of four spatial layers, white matter layer (WML), granule

A

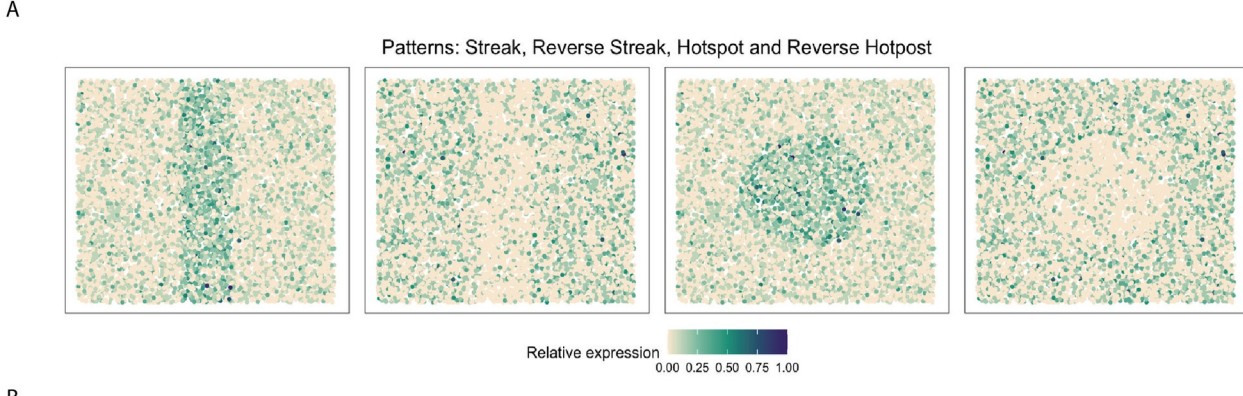

B

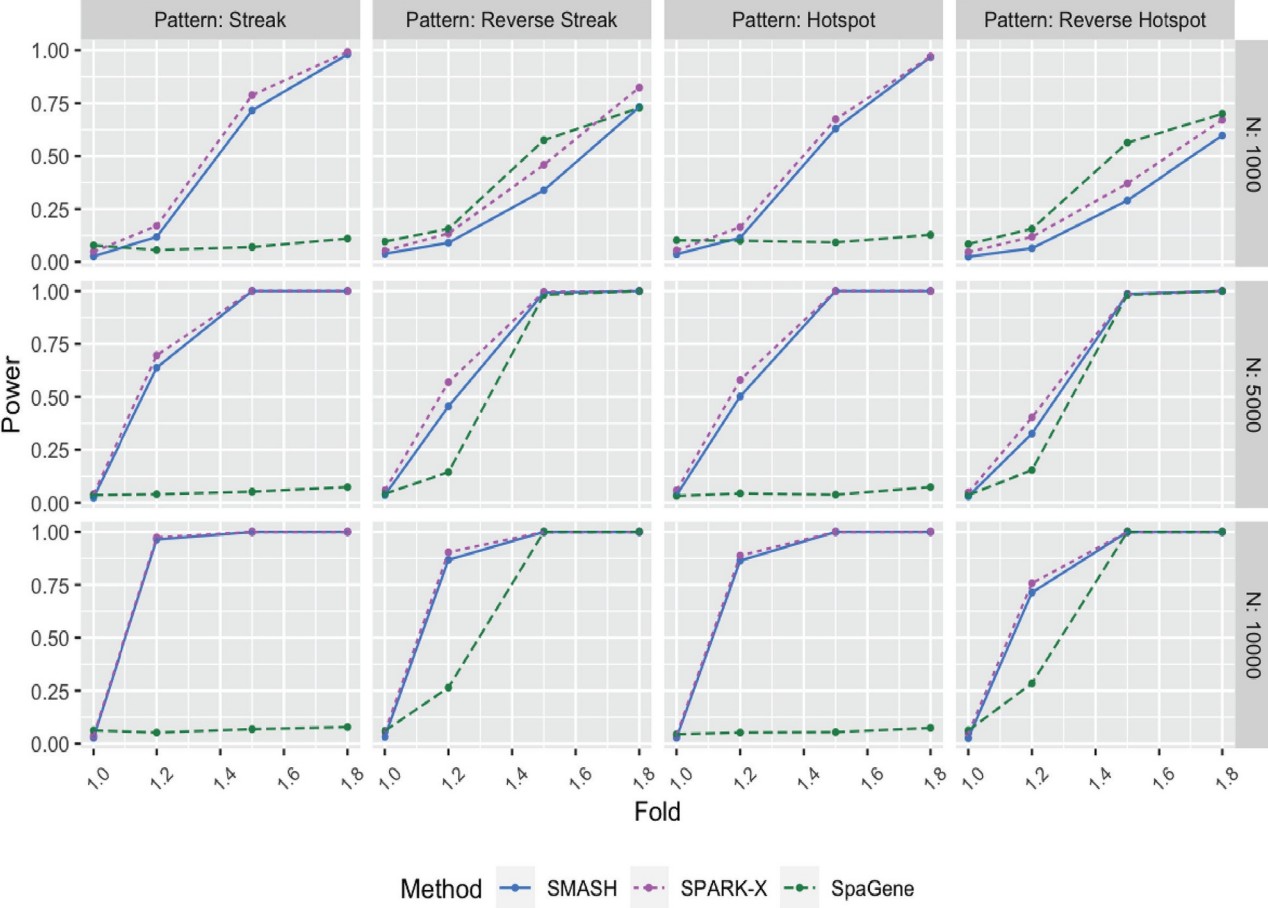

**Fig 1. Simulation following the SPARK-X manuscript. A** Four spatial expression patterns that the genes were assumed to follow. **B** Statistical power plots of the three methods, SMASH, SPARK-X, and SpaGene under varying values of $N$ and fold-size, for $K = 500$ genes at a level of $\alpha = 0.05$. The results were averaged over five replications.

layer (GL), Purkinje layer (PL), and molecular layer (ML) [62]. These layers consist of different types of cells. For example, WML contains oligodendrocytes, GL contains granule cells, PL contains Purkinje neurons and Bergmann gila, and ML contains intra-neurons MLI. These cell types can be inferred based on just the transcriptional profiles using cell clustering software

A

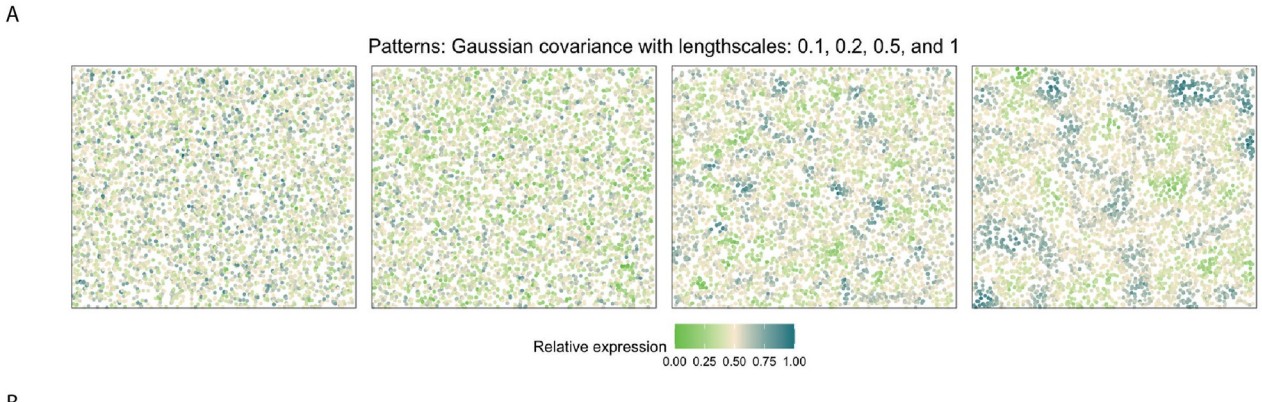

Patterns: Gaussian covariance with lengthscales: 0.1, 0.2, 0.5, and 1

B

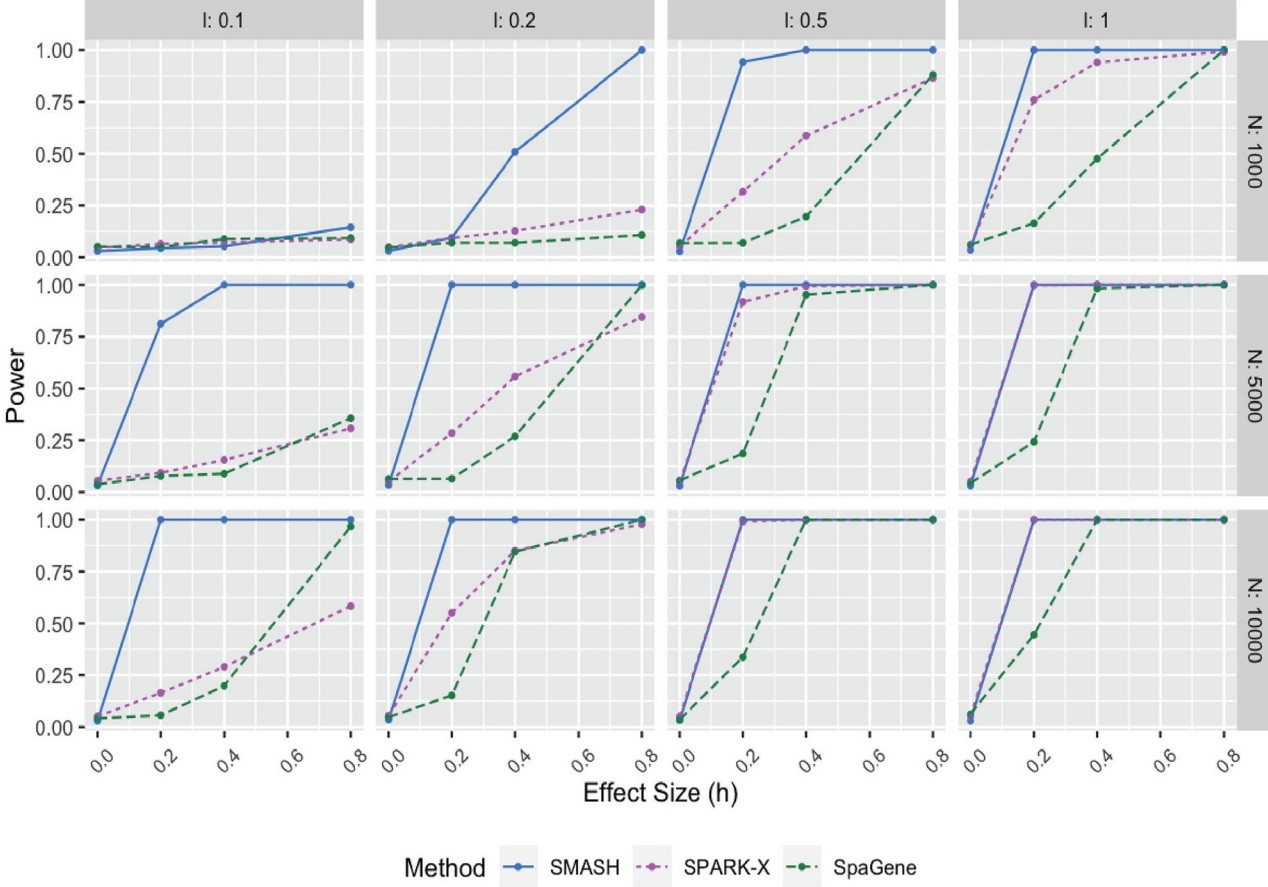

**Fig 2. Simulation using Gaussian process-based regression model with the Gaussian covariance.** A) Four spatial expression patterns that were generated using Gaussian covariance matrices with four different values of the lengthscale *l*. B) Statistical power plots of the three methods under varying values of *N* and effect-size (*h*) for *K* = 1000 genes at a level of $\alpha$ = 0.05. The results were averaged over five replications.

like RCTD [32]. We display the inferred cell types overlayed on the spatial locations in Fig 4. Out of the 7,653 genes, SMASH identified 1173 genes to be spatially variable (adjusted *p*-value: $p_{\text{adjust}} < 0.05$). SPARK-X and SpaGene respectively detected 608 and 518 genes, and the overlaps between the detected SVGs by the three methods are displayed in a Venn diagram (Fig 4).

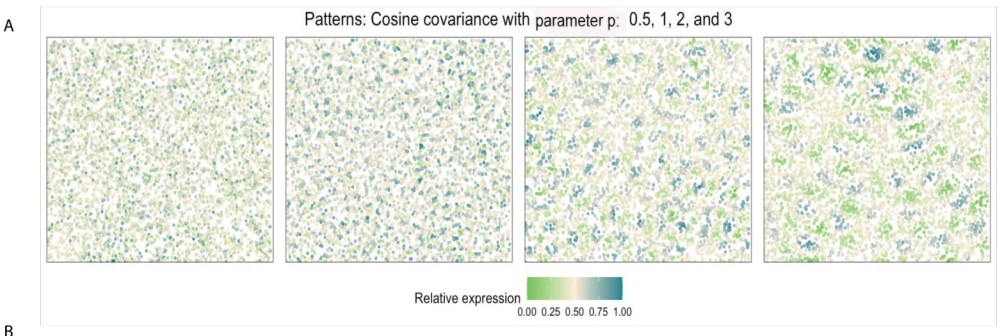

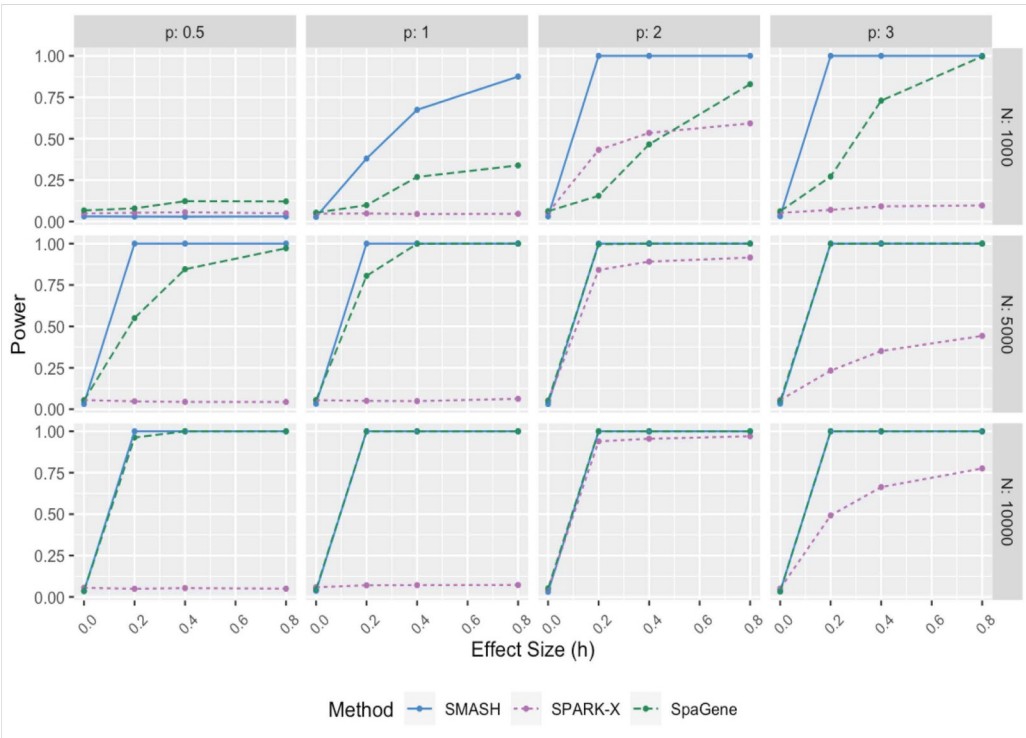

**Fig 3. Simulation using Gaussian process-based regression model with the cosine covariance.** A) Four spatial expression patterns that were generated using cosine covariance matrices with four different values of the period $p$. B) Statistical power plots of the three methods under varying values of $N$ and effect-size ($h$) for $K = 1000$ genes at a level of $\alpha = 0.05$. The results were averaged over five replications.

**Table 1. Computational complexity and run-time comparison.** The table lists the theoretical complexity and run-time (in seconds) of the four methods, SMASH, SPARK-X, SpaGene, and SpatialDE in a simulation setup with $K = 1000$ genes and varying number of cells $N$. The number of spatial coordinates $d$ was equal to 2. *SpaGene constructs multiple kNN graphs and performs permutation tests. We are only listing the complexity of the KNN algorithm.

| Method | Complexity | N = 1000 | N = 5000 | N = 10000 |
|---|---|---|---|---|
| SMASH | $O(N^2K)$ | 0.9 | 18.5 | 97.5 |
| SPARK-X | $O(NKd^2)$ | 0.72 | 4.1 | 4.3 |
| SpaGene* | $O(NKd)^*$ | 0.2 | 1.1 | 2.1 |
| SpatialDE | $O(N^3 + N^2K)$ | 24.5 | 245 | 971.7 |

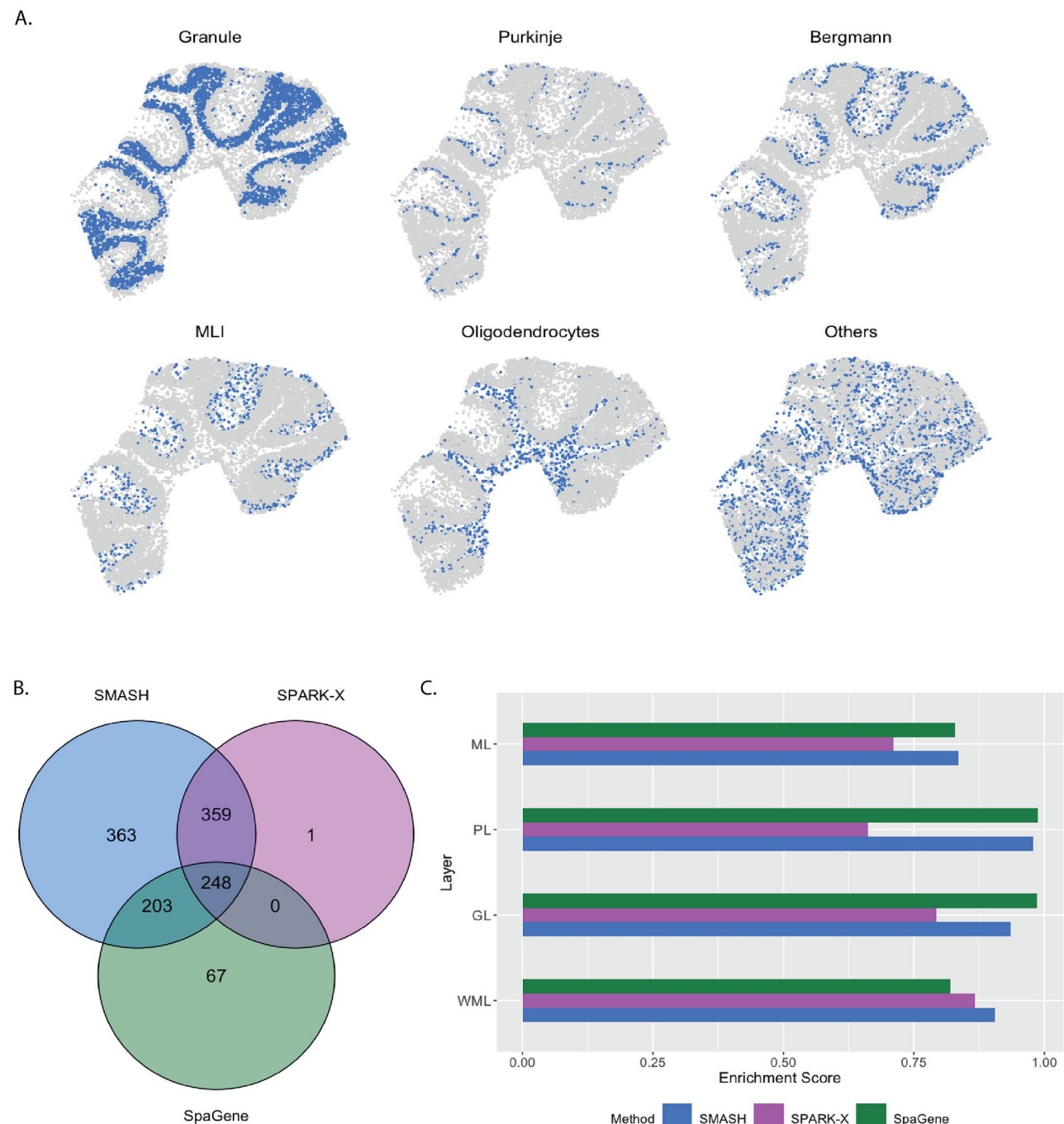

**Fig 4. Analysis of mouse cerebellum data.** A) Location of the major cell types corresponding to the four spatial layers of the mouse cerebellum. B) Overlap between the detected SVGs by the three methods. C) Enrichment scores of the methods in the four spatial layers.

We noted that SPARK-X and SpaGene had many of the SVGs uncommon. SMASH, on the other hand, could identify almost all the detected genes by those two methods, especially SPARK-X, while detecting an additional 363 SVGs.

Next, we performed two types of enrichment analysis. First, we compared the performance of the methods in different layers by computing their enrichment scores (ES) following Liu

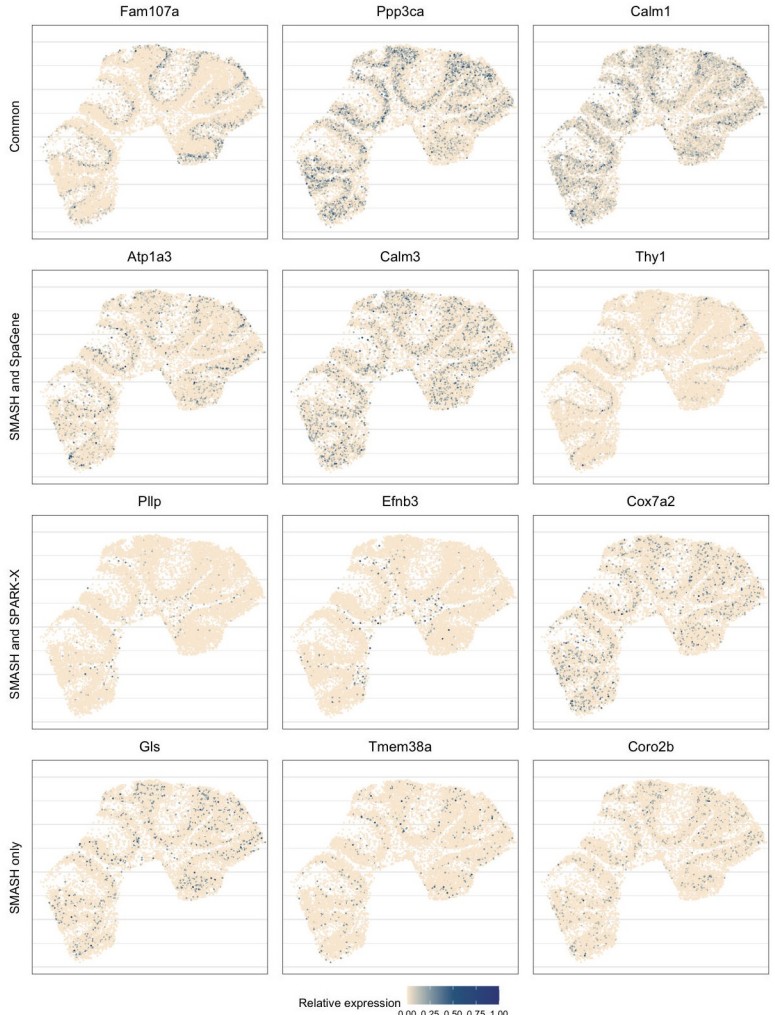

**Fig 5. Expression patterns in mouse cerebellum data.** Three representative genes from the detected pathways for the four sets of genes: a) the common genes identified by all three methods, b) the genes identified by SMASH and SpaGene but not by SPARK-X, c) the genes identified by SMASH and SPARK-X but not by SpaGene, and d) the genes identified only by SMASH.

et al. (2022) [48]. It is based on the expectation that the genes which abundantly express themselves in the four spatial layers, should be identified and ranked top by the methods. In that regard, we noticed that SPARK-X performed poorly in the PL, whereas SpaGene performed poorly in the WML. SMASH, on the other hand, consistently achieved similar or better performance compared to the other two methods in all four layers. Secondly, we performed functional enrichment analysis of the following four sets of SVGs: a) the common genes identified by all three methods, b) the genes identified by SMASH and SpaGene but not by SPARK-X, c) the genes identified by SMASH and SPARK-X but not by SpaGene, and d) the genes identified only by SMASH. The expression pattern of three representative genes of the enriched pathways for each of these four sets of genes, are shown in Fig 5. For set (a), top enriched Gene Ontology (GO) terms, such as GO: 0098916 (anterograde trans-synaptic signaling), GO: 0007268 (chemical synaptic transmission), and GO: 0099536 (synaptic signaling), were broadly associated with synaptic regulation. The protein-coding genes Fam107a, Ppp3ca, and Calm1 appeared in these top pathways. Fam107a seems to express in the PL, whereas the other

two express in the GL (Fig 5). For set (b), the top GO terms including GO: 0006873 (intracellular monoatomic ion homeostasis), GO: 0030003 (intracellular monoatomic cation homeostasis), and GO: 0098771 (inorganic ion homeostasis) were associated with ion homeostasis. The representative genes Atp1a3 and Thy1 express in the PL while Calm3 expresses in the GL. For set (c), the top pathways including GO: 0006811 (monoatomic ion transport), GO: 0006812 (monoatomic cation transport), and GO: 0098655 (monoatomic cation transmembrane transport) were associated with ion transportation. The representative genes Pllp and Efnb3 express in the WML, whereas Cox7a2 expresses roughly in the GL. For set (d), the top enriched GO terms, such as GO: 0044057 (regulation of system process) and GO: 0050877 (nervous system process), were associated with regulating different types of system processes. The representative genes Gls, Tmem36a, and Coro2b roughly express in the GL.

**Human DLPFC by Visium.**  The human dorsolateral prefrontal cortex (DLPFC) data [11] has 33,538 and 3,639 spots. We focused on the 13,783 genes which express in more than 1% of the spots. Every spot belongs to one of the six manually labeled cortical layers or the white matter layer (WML) (Fig 6). SMASH and SPARK-X identified 10,871 and 10,416 SVGs respectively ($p_{\text{adjust}} < 0.05$), whereas SpaGene identified only 2379. The overlaps between the detected SVGs by the three methods are displayed in a Venn diagram (Fig 6). We noted that almost all the genes detected by SpaGene were also detected by both SMASH and SPARK-X. SMASH and SPARK-X detected a lot of additional SVGs. We performed functional enrichment analysis of the two sets of detected genes: a) the common genes identified by all three methods and b) the genes identified only by SMASH and SPARK-X but not by SpaGene. For set (a), top enriched GO terms, such as GO: 0099537 (trans-synaptic signaling) and GO: 0099177 (regulation of trans-synaptic signaling), were associated with synaptic signaling. For set (b), top enriched GO terms like GO: 0006397 (mRNA processing) and GO: 0000375 (RNA splicing, via transesterification reactions), were associated with RNA processing. The expression of three representative genes from the set (b) are displayed in Fig 6. There seemed to be a gradient spatial pattern of expression for all three genes which SpaGene failed to detect. Similar to the previous section, we computed the enrichment score (ES) of every method in the seven manually labeled spatial layers. From Fig 6, we noticed that SpaGene performed poorly in terms of ES, especially in Layers 1 and 6. We also performed an additional check as follows. There are three cortical-layer associated SVGs, MOBP, SNAP25, and PCP4, and three blood and immune-related SVGs, HBB, IGKC, and NPY, known to be spatially variable from previous studies [11]. We checked how many of these genes appeared in the lists of the top thousand SVGs (in terms of $p_{\text{adjust}}$) by the three methods. SMASH and SpaGene respectively ranked five and six of these SVGs, whereas SPARK-X ranked only two cortical-layer associated genes.

**SCCOHT by Visium.**  The small cell carcinoma of the ovary hypercalcemic type (SCCOHT) data [63] has 15,229 genes and 2071 cells. We restricted our focus to the 12,001 genes that express in more than 5% of the cells. Sanders et al. (2022) [63] grouped the cells into twelve clusters based on the expression profile of a selected few genes, using Seurat [33], which we display in Fig 7. SMASH, SPARK-X, and SpaGene respectively detected 9361, 6564, and 6899 SVGs ($p_{\text{adjust}} < 0.05$). The overlaps between the detected SVGs by the three methods are displayed in a Venn diagram (Fig 7). SMASH could detect most of the SVGs identified by at least one of the other two methods and an additional 1634 genes. Similar to the analysis of the mouse cerebellum data, we checked if the methods could identify the top genes that show enriched expression in the twelve spatially well-separated clusters found by Sanders et al. (2022). We computed the enrichment scores (ES) of the methods for each of the clusters (Fig 7). SMASH achieved consistently higher ES for all the clusters while SpaGene was the second best in most cases. Additionally, in Fig 8, we show the expression of three chosen genes from each of the following four sets of SVGs, a) the common genes identified by all three methods,

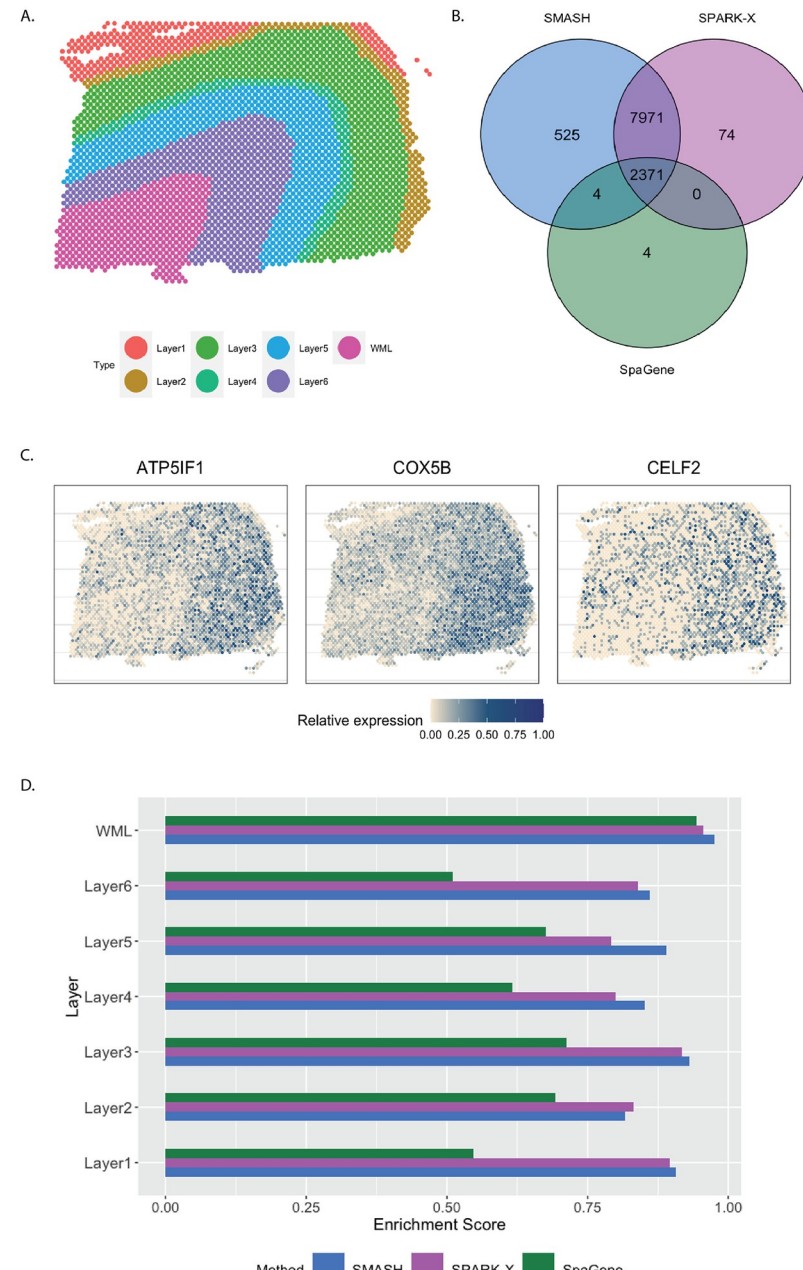

**Fig 6. Analysis of human DLPFC data.** A) Manually labeled cortical layers (layers 1–6) and white matter layer (WML). B) Overlap between the detected SVGs by the three methods. C) Expression of three representative genes identified only by SMASH and SPARK-X. D) Enrichment scores of the methods in different layers.

b) the genes identified by SMASH and SpaGene but not by SPARK-X, c) the genes identified by SMASH and SPARK-X but not by SpaGene, and d) the genes identified only by SMASH. We also checked the clinical relevance of these genes in the existing literature. For example, CITED4, which was detected to be an SVG by all three methods, has been found to be associated with lung adenocarcinoma [64]. From the set (b), ELF4A1 has been found to be associated with gastric cancer [65]. EZH2, from the set (c), is a well-known marker for being associated

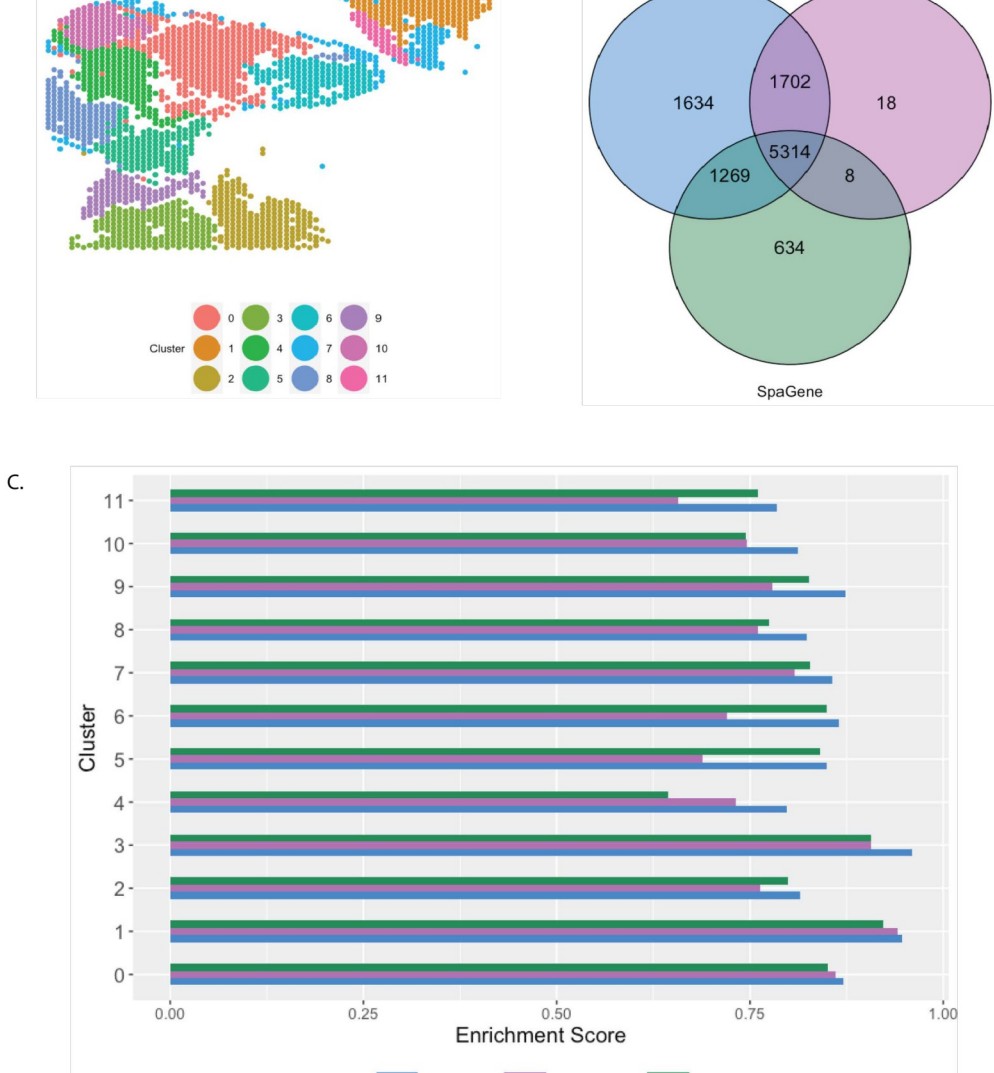

**Fig 7. Analysis of SCCOHT data.** A) Pre-identified clusters of cells using Seurat. B) Overlap between the detected SVGs by the three methods. C) Enrichment scores of the methods in different clusters.

with the development and progression of different types of cancer [66, 67]. Sanders et al. (2022) [63] also found the expression of EZH2 to be highly variable across their identified spatial clusters. Finally, from the set (d), SEMA4F has been found to be associated with endometrial cancer [68].

**Mouse hypothalamus by MERFISH.** The mouse hypothalamus data [60] has 161 genes and 5665 cells. 156 genes are pre-selected markers for different cell types and can thus be expected to be highly variable, whereas the other five are control genes. The cell types, such as endothelial, ependymal, and inhibitory, can be identified based on the transcriptional profiles of the markers. The spatial organizations of a few major cell types are shown in Fig 9.

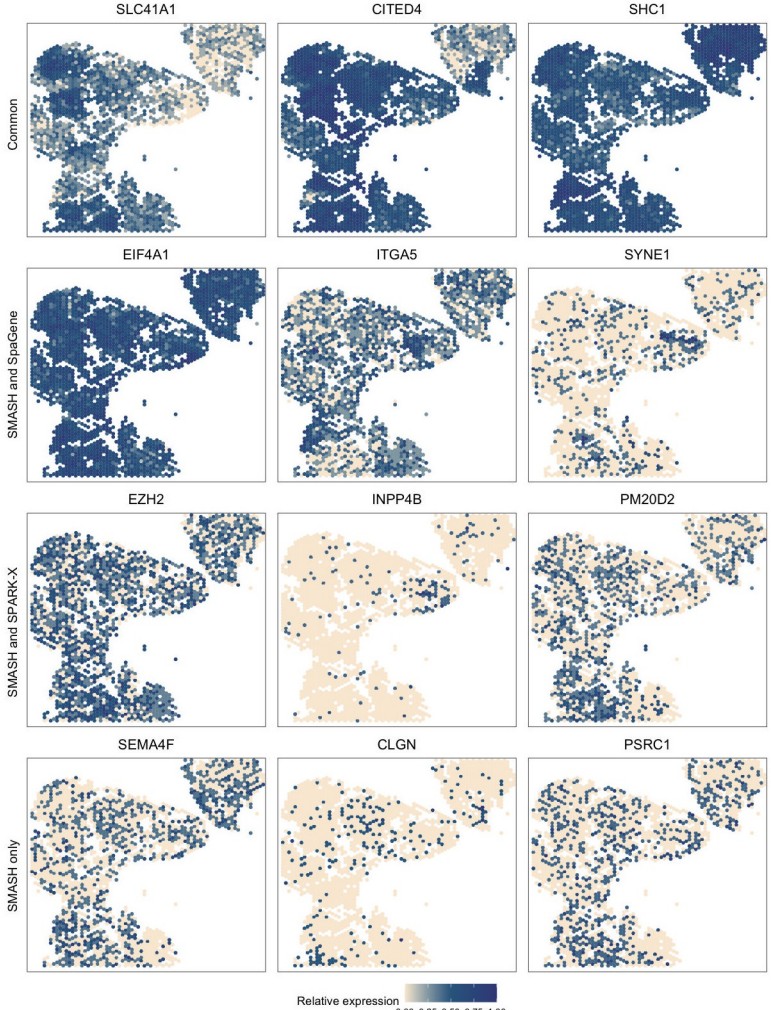

**Fig 8. Expression patterns in SCCOHT data.** Three representative genes from the four sets of SVGs: a) the common genes identified by all three methods, b) the genes identified by SMASH and SpaGene but not by SPARK-X, c) the genes identified by SMASH and SPARK-X but not by SpaGene, and d) the genes identified only by SMASH.

SMASH was able to detect 139 genes, whereas SPARK-X and SpaGene detected 127 and 124 genes, respectively ($p_{adjust} < 0.01$). The overlaps between the SVGs detected by the three methods are shown in Fig 9. SMASH identified all the SVGs SPARK-X could detect, while SpaGene identified one additional SVG. It should be highlighted that all the methods assigned the five control genes to not be spatially variable. We display the expression of two representative genes from three sets of genes, a) the genes identified only by SMASH and SpaGene, b) the genes identified only by SMASH and SPARK-X, and c) the genes identified only by SMASH. We did not focus on the common genes because they have been extensively studied in earlier literature, such as the work of Liu. et al. (2022) [48]. The genes Npy1r and Cplx3 belonged to set (a), and are known to be enriched in inhibitory and excitatory neurons [69, 70]. Rxfp1 and Ntsr1 belonged to set b). Even though both genes are known to express in inhibitory and excitatory neurons, Rxfp1 seems to express in ependymal cells as well. Galr2 and Crhr1 are two genes from set c) which express in multiple cell types including inhibitory cells and astrocytes.

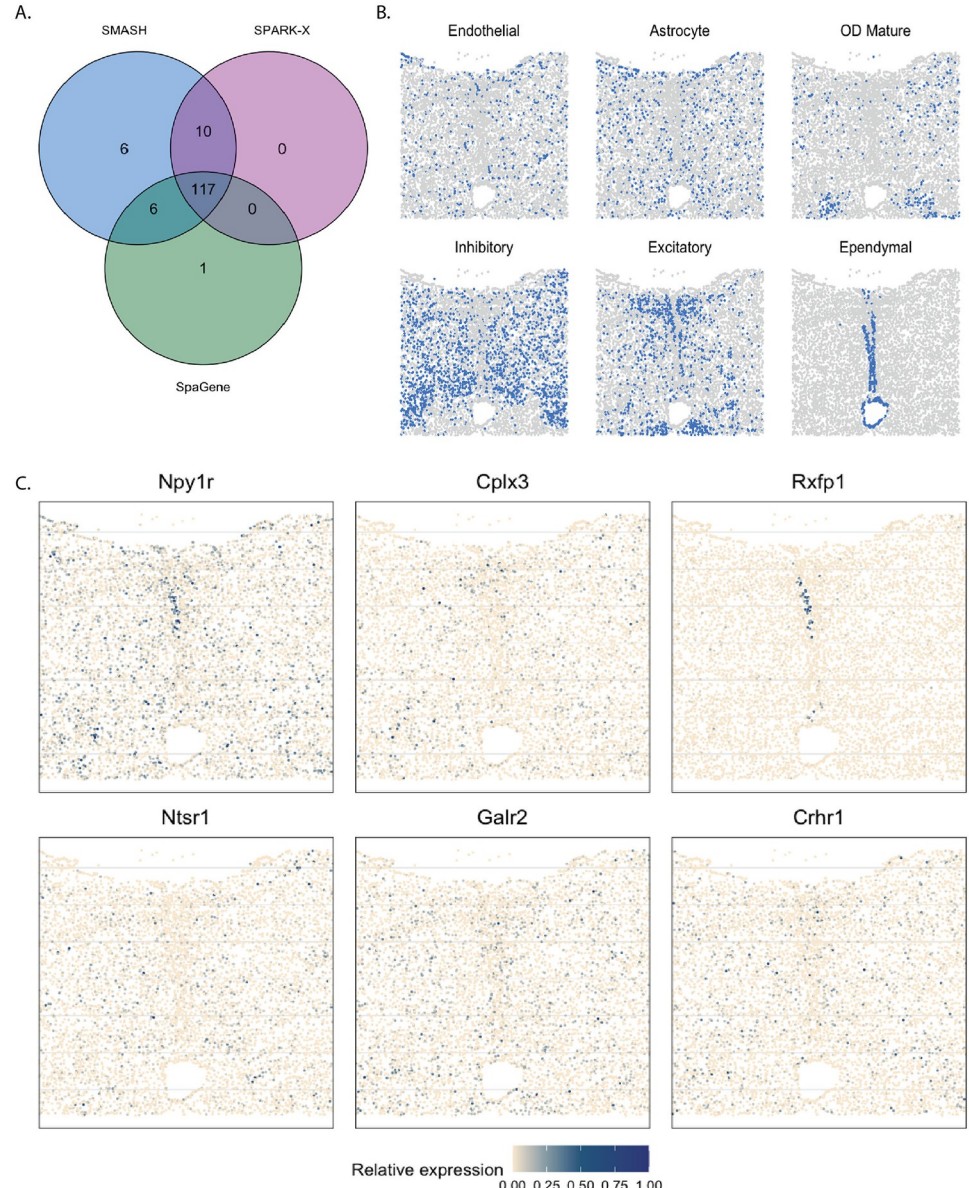

**Fig 9. Analysis of mouse hypothalamus data.** A) Overlap between the detected SVGs by the three methods. B) Spatial organization of a few major cell types. C) Expression of two representative genes from each of the three sets, a) the genes identified only by SMASH and SpaGene, b) the genes identified only by SMASH and SPARK-X, and c) the genes identified only by SMASH.

## Discussion

We have proposed a novel non-parametric method SMASH for detecting spatially variable genes (SVGs) in the context of large-scale spatial transcriptomics (ST) datasets. In comparison to existing scalable approaches, SMASH achieves superior power in both complex simulation scenarios and real data analyses while remaining computationally tractable.

Recently developed spatial transcriptomics platforms produce high-dimensional datasets [18–20] in terms of the number of cells and the number of genes. In such large datasets, fully parametric approaches for detecting SVGs, such as SpatialDE [34] and SPARK [37], albeit

statistically powerful, become intractable for their high computational demand. Computationally efficient alternative non-parametric approaches, such as SPARK-X [38] and SpaGene [35], on the other hand, can often turn out to be significantly less powerful. In our method SMASH, we strive to find a balance between these two issues, achieving higher statistical power while attaining a moderate degree of scalability. We augment the kernel-based covariance testing framework [54], used before in SPARK-X, by accounting for more complex spatial dependencies.

In three different simulation setups, one following the SPARK-X manuscript [38] and the other two following the framework of SpatialDE [34], we evaluated the performance of SMASH, along with two other methods: SPARK-X and SpaGene, in terms of type 1 error and power. SMASH achieved consistently similar or better power than the other two methods in all the simulation setups for all combinations of the varying parameters. In contrast, both SPARK-X and SpaGene behaved unpredictably, achieving almost zero detection power in many of the cases. It demonstrated their lack of robustness and failure to capture complicated structures of spatial dependency in the gene expression. In the run-time comparison of the methods, we showed that SMASH, although slower than SPARK-X and SpaGene, remained fairly tractable and was almost ten times faster than a fully parametric approach like SpatialDE. SMASH, SPARK-X, and SpaGene were then applied to four real datasets: 1) mouse cerebellum data collected using Slide-seq V2 [19], 2) human dorsolateral prefrontal cortex data collected using Visium [11], 3) small cell ovarian carcinoma of the ovary hypercalcemic type data collected using Visium [11], and 4) mouse hypothalamus data collected using MERFISH [60]. We compared the methods via a number of avenues: a) checking the overlap between the detected SVGs by the three methods, b) computing enrichment scores (ES) of the methods in different spatial layers or cell types identified based on the transcriptional profiles using popular softwares, such as RCTD [32] and Seurat [33], and c) investigating the functional enrichment of the genes that were detected by SMASH but remained undetected by at least one of the other two methods. For all the datasets, SMASH detected more SVGs than the other two methods, which included nearly all of the SVGs detected by SPARK-X. SMASH could also detect most of the SVGs that were identified by SpaGene but not by SPARK-X. For example, in data (1), from the 7,653 genes after quality control, SMASH identified 1173 SVGs which included 607 out of the 608 SVGs SPARK-X could detect. Out of the 518 SVGs detected by SpaGene, only 248 were also detected by SPARK-X, while SMASH detected 451 of them. It is important to highlight that SMASH produced calibrated $p$-values in the null simulations from all of these datasets, lending credibility to these higher numbers of detected SVGs. In the same dataset, SMASH achieved a higher enrichment score (ES) than the other two methods in different pre-identified spatially separated layers or cell types of the mouse cerebellum. A higher ES implied better capability to identify the genes that showed highly variable expression in a particular spatial layer compared to the rest. In the other datasets as well, SMASH consistently achieved better ES in different spatially localized cell types. We also studied the functional properties and clinical significance of the identified SVGs. For example, in data (3), the gene EZH2 was detected to be spatially variable by SMASH and SPARK-X. EZH2 is a known marker for the progression of different types of cancers [66, 67].

In all the methods we have discussed, including SMASH, the biology of a single tissue section from a single subject is explored at a time. It means that if we either have multiple tissue sections from the same subject or from multiple subjects, the methods will have to identify SVGs individually, disregarding the shared information between and across the subjects. Thus, we would like to extend SMASH in a hierarchical fashion for jointly analyzing more than one tissue section or subject in the future. One more important functionality that we would like to incorporate would be the ability to classify the genes based on their similarity of

spatial expression patterns. For example, SpatialDE [34] considers a hierarchical Bayesian mixture model approach that suffers from extremely high computational demand. SpaGene [48] considers a non-negative matrix factorization [71] of the expression data to identify similarly expressed genes. This approach, although computationally feasible, does not take into account the spatial locations directly and can thus be suboptimal in capturing truly spatial patterns. In the future, we would like to study this problem with a deeper focus and pursue methodological development in this area. Finally, we would like to explore the possibility of using SMASH in the context of multiplex immunohistochemistry (mIHC) datasets [72, 73] where the goal is to identify spatially variable cell types and their interaction.

## Materials and methods

We briefly discuss some of the existing methods such as SpatialDE [34], SPARK [37], SPARK-X [38], and SpaGene [35], and then present the proposed method SMASH. Note that we did not compare SMASH to either SpatialDE or SPARK in our Results section except for the time comparison, primarily due to their high computational demand and the fact that these have already been studied in great detail in earlier works. However, we still discuss their modeling frameworks to facilitate comparisons. Let us introduce a few relevant notations. Suppose there is a single subject (image) with $N$ cells/spots and the expression profile of $K$ genes is observed in the cells. For the $i$-th cell, let $s_i$ denote its location i.e., a vector of spatial (two or three-dimensional) coordinates, and $y_{ki}$ denote the expression of the $k$-th gene in the cell. Let us also define, $y_k = (y_{k1}, \ldots, y_{kN})^T$ and $S = (s_1, \ldots, s_N)^T$. For the sake of simplicity, we are assuming that there are no additional covariates but in all the methods, except SpaGene, covariates can be readily incorporated.

### A brief overview of existing methods

**SpatialDE.** SpatialDE uses a Gaussian process (GP)-based spatial regression model [49, 74]. which has the following form in a finite sample,

$$y_k \sim N(\mu_k \mathbf{1}, \tau_k^2 \Sigma + \sigma_k^2 I)); \quad \Sigma = [[\Sigma_{ij}]]_{N \times N}; \quad \Sigma_{ij} = \exp\left[-\frac{||s_i - s_j||^2}{2l^2}\right]; \quad (1)$$

where $\mathbf{1}$ denotes the $n$-length vector of all 1's, $I$ denotes the $N$-dimensional identity matrix and $\Sigma$ denotes a Gaussian covariance matrix. $||.||$ denotes the Euclidean norm, and the hyperparameter $l$, known as the characteristic lengthscale [75, 76], controls the rapidness at which the covariance decays as a function of the spatial distance. The fixed effect $\mu_k$ accounts for the mean expression level and $\tau_k^2$ accounts for the expression variance attributable to spatial effects. A large value of $\tau_k^2$ should imply that the gene shows differential spatial expression. To formally test the hypothesis, $H_0 : \tau_k^2 = 0$ against $H_1 : \tau_k^2 > 0$, SpatialDE considers the likelihood ratio test (LRT) [77]. To estimate the model parameters under the full model, the log-likelihood corresponding to Eq (1) is optimized w.r.t. $(\mu_k, \tau_k^2, \sigma_k^2)$ using an efficient algorithm by Lippert et al. (2011) [51]. Ideally, it is desirable to optimize over the hyperparameter $l$ as well but for the sake of computational feasibility, $l$ is kept fixed at a few carefully chosen values. For every choice of $\Sigma$, to analyze all $K$ genes, the efficient algorithm requires just one computationally demanding step with a complexity of $O(N^3)$, instead of $O(N^3 K)$ as incurred in naive algorithms. Along with the Gaussian covariance function, SpatialDE also considers linear and cosine covariance functions to construct $\Sigma$, and finally, combines all the LRT values corresponding to different choices of $\Sigma$ for the inference. For a particular $\Sigma$, the computational complexity of SpatialDE is of $O(N^3 + N^2 K)$.

**SPARK and SPARK-X.** SPARK [37] extends Eq 1 by considering a generalized linear spatial model (GLSM) [52] with Poisson distribution as

$$y_k(s_i) \sim Poi(N(s_i)\lambda_k(s_i)); \quad (\log(\lambda_k(s_1)), \ldots, \log(\lambda_k(s_N)))^T \sim N(\mu_k \mathbf{1}, \tau_k^2 \Sigma + \sigma_k^2 I). \quad (2)$$

For cell $i$, $\lambda_k(s_i)$ is an unknown Poisson rate parameter that represents the underlying gene expression. The variance parameters, $\tau_k^2$ and $\sigma_k^2$ have similar interpretations as earlier. To test $H_0 : \tau_k^2 = 0$, SPARK uses the score test [78]. Parameter estimation and inference are incredibly hard in GLSM which is why SPARK uses an approximate algorithm based on the penalized quasi-likelihood (PQL) approach [53, 79]. The approach has the computational complexity of $O(N^3)$ for every trait, or $O(N^3 K)$ in total. Thus, it lacks severely in terms of scalability.

Improving upon SPARK's scalability, a recent non-parametric method named SPARK-X [38] has been proposed. The method is built on a simple intuition: if $y_k$ is independent of $S$, the spatial distance between two locations $i$ and $j$ should be independent of the difference in gene expression between the two locations. It computes the expression covariance matrix, $E_k = y_k(y_k^T y_k)^{-1} y_k^T$ and the distance covariance matrix, $D = S(S^T S)^{-1} S^T$ and constructs the test statistic as, $T_k^{\text{SPARKX}} \equiv tr(E_k D)/N$ where $tr()$ denotes the trace operator. Assume $y_k$ to be mean-standardized for the sake of simplicity. Under the null hypothesis of no association, $T_k$ asymptotically follows a weighted mixture of independent $\chi_1^2$ distributions. The weights are the products of the ordered eigenvalues of the matrices, $E_k$, and $D$. SPARK-X requires the computational complexity of just $O(Nd^2)$ for every gene, or $O(NKd^2)$ in total, where $d$ is the dimension of the location-space $\mathcal{S}$, e.g., $d = 3$ if $\mathcal{S} = \mathbb{R}^3$. A linear complexity w.r.t. $N$ makes SPARK-X easily applicable to large-scale ST datasets. SPARK-X also considers several element-wise non-linear transformations of $S$ as $g(S)$, where $g$ is a Gaussian or cosine transformation (not to be confused with Gaussian or cosine kernels), and repeats the above testing procedure replacing $S$ with $g(S)$. The $p$-values are combined using a Cauchy $p$-value combination rule [80].

However, the form of $D$ corresponds to a linear covariance function [75]. It makes SPARK-X equivalent to performing a multiple linear regression of $y_k$ on $S$ or $g(S)$ and testing if the fixed effect parameters differ from zero. Thus, SPARK-X is only capable of detecting first-order spatial dependencies and as shown in the Results section, severely lacks power for higher-order dependencies.

**SpaGene.** A very recently developed method, SpaGene [48], is different from the rest of the methods discussed so far in the sense of being model-free and based on graphs. The intuition behind the method is that the cells/spots with high gene expression are more likely to be spatially connected than random. It constructs the $k$-nearest neighbor (kNN) graph based on spatial locations. Then, for each gene, it extracts a subnetwork comprising only cells/spots with high expression from the kNN graph. SpaGene quantifies the connectivity of the subnetwork using the earth mover's distance (EMD) [56] between degree distributions of the subnetwork and a fully connected one. To generate the null distribution of the EMD for inference, a permutation test is considered. For further details, we refer the readers to the original manuscript [41].

## Proposed method: SMASH

**Setup.** We test the null hypothesis of $y_k$ and $S$ being independent, i.e., $H_0: y_k \perp S$, using a non-parametric kernel-based framework [58, 81–83]. Let $y_k$ and $S$ have domains $\mathcal{Y}$ and $\mathcal{S}$, respectively. Denote $k_{\mathcal{Y}}$ and $k_{\mathcal{S}}$ to be two measurable positive definite (PD) kernels with the corresponding reproducible kernel Hilbert spaces (RKHSs) denoted by $H_{\mathcal{Y}}$ and $H_{\mathcal{S}}$ on $\mathcal{Y}$ and

$S$, respectively. Then, the cross-covariance operator: $\Sigma_{Sy}$ from $H_y$ to $H_S$ can be defined by the relation: $< f_1, \Sigma_{Sy} f_2 >_{H_S} = cov(f_2(y_k), f_1(S)), \forall f_1 \in H_S, f_2 \in H_y$, where $<.>$ denotes an inner product. $\Sigma_{Sy}$ can be interpreted as a more general version of the covariance matrix on Euclidean spaces, representing higher-order correlations of $y_k$ and $S$ through $f_2(y_k)$ and $f_1(S)$. Under additional regulatory assumptions on RKHSs: $H_y$ and $H_S$ [58], it can be shown that testing $H_0$: $y_k \perp\!\!\!\perp S$ is equivalent to testing, $H_0 : \Sigma_{Sy} = 0$. This testing can be performed using a test statistic of the form $tr(K_y K_S)$, where $K_y$ and $K_S$ are the kernel covariance or Gram matrices, obtained using the PD kernels: $k_y$ and $k_S$ [75, 76]. In the context of real datasets, exact choices for $k_y$ and $k_S$ are never known. Therefore, we consider different kernel choices and aggregate the results. Our test statistic has the form $T_k^{\text{SMASH}} \equiv tr(E_k K_S)/N$, where $E_k$ is defined as earlier, i.e., $k_y$ is fixed to be a linear kernel, while $k_S$ and consequently, $K_S$ is varied to have different forms as described next.

**Kernels and hyperparameters.** In this work, we consider $K_S$ to have three forms: a) the Gaussian kernel covariance matrix, $\Sigma$ defined in Eq (1), b) a cosine or periodic kernel covariance matrix of the form, $K_S = [[\cos(2\pi||s_i - s_j||/p)]]_{N \times N}$, where parameter $p$ is known as the period, and c) the linear kernel-based covariance matrix $D$ considered in SPARK-X (with Gaussian and cosine transformations of $S$ as well). For the Gaussian and cosine covariance matrices, we consider ten data-driven fixed values of the lengthscale $l$ and period $p$, respectively (see S1 Text). Refer to Fig A in S1 Text, for visualizing the spatial patterns corresponding to the different kernel covariance matrices. In Table 2, we list the kernel covariance matrices used in different methods. Note that $T_k^{\text{SPARKX}}$ can be interpreted as a special case of $T_k^{\text{SMASH}}$ as the former only considers linear kernel covariance matrices.

**Distribution and computational complexity.** For a particular choice of $K_S$, the asymptotic null distribution of $T_k^{\text{SMASH}}$ is a weighted mixture of independent $\chi_1^2$ distributions, where the weights are the products of the ordered eigenvalues of the matrices, $E_k$, and $K_S$ [54, 58]. However, unlike the kernel choices of $T_k^{\text{SPARKX}}$, $K_S$ does not always have a projection matrix-like structure as $D$, and thus, its eigenvalues can not be computed with the complexity of $O(Nd^2)$. Instead, it requires the complexity of $O(N^3)$, rendering it intractable as $N$ increases. Therefore, we consider a variation of Welch-Satterthwaite approximation [84, 85], to approximate the asymptotic null distribution of $T_k^{\text{SMASH}}$ with a gamma distribution [58] as below,

$$T_k^{\text{SMASH}} \sim \Gamma(\theta_1, \theta_2); \quad \theta_1 = \frac{\mathbb{E}(T_k^{\text{SMASH}})^2}{\mathbb{V}(T_k^{\text{SMASH}})}; \quad \theta_2 = \frac{\mathbb{V}(T_k^{\text{SMASH}})}{\mathbb{E}(T_k^{\text{SMASH}})};$$

$$\mathbb{E}(T_k^{\text{SMASH}}) = \frac{1}{N^2} tr(E_k) tr(K_S); \quad \mathbb{V}(T_k^{\text{SMASH}}) = \frac{2}{N^4} tr(E_k^2) tr(K_S^2);$$

where $\mathbb{E}()$ and $\mathbb{V}()$ denote the expectation and variance, respectively. It is easy to verify that $tr(E_k) = tr(E_k^2) = 1$. Notice that we can now avoid any operation of complexity $O(N^3)$. Computation of $tr(K_S^2)$ just requires the complexity of $O(N^2)$ using the property that $tr(AB) = \sum_{i=1}^{N} \sum_{j=1}^{N} a_{ij} b_{ij}$ for two matrices, $A = [[a_{ij}]]_{N \times N}$ and $B = [[b_{ij}]]_{N \times N}$ [86]. Thus, for a particular

**Table 2. Kernel choices in different methods.** The table shows (yes/no) if a particular kernel covariance or Gram matrix is considered in different methods.

| Method | Linear | Gaussian | Cosine | Linear with transformed coordinates |
|---|---|---|---|---|
| SMASH | Yes | Yes | Yes | Yes |
| SPARK-X | Yes | No | No | Yes |
| SpatialDE | Yes | Yes | Yes | No |
| SPARK-X | Yes | Yes | Yes | No |

choice of $K_S$, to analyze all $K$ genes, SMASH requires the complexity of $O(N^2K)$. This computational complexity is higher than SPARK-X. But we are making that sacrifice to gain significantly more power, as shown in both simulation studies and real data analyses while still achieving a moderate degree of scalability. It is worth pointing out that even though SMASH is non-parametric and does not make any distributional assumptions, $T_k^{\text{SMASH}}$ shares a close similarity with the SpatialDE model under some additional assumptions (see S1 Text).

**Aggregation and covariates.** As mentioned earlier, we consider multiple (say, $R$) choices for $K_S$, to construct multiple test statistics: $T_{kr}^{\text{SMASH}}$, $r = 1, \ldots, R$. Finally, we combine the $p$-values corresponding to these test statistics using the minimum $p$-value combination rule [80] (see S1 Text for more details). Note that we have assumed that $y_k$ is mean-standardized and there are no additional covariates to be taken into account. In the presence of covariates, we would regress the covariates out from the gene expression vector $y_k$, prior to performing the test, using a multiple linear regression model. To further elaborate, letting $X$ be the corresponding matrix of covariates, we would compute the projection matrix $P_X = X(X^TX)^{-1}X^T$, and substitute the vector $y_k$ with $y_k^* = [I - P_X]y_k$, in our proposed test statistic.

## FDR control and non-PD kernels

In the real data analysis, we used Benjamini-Yekutieli [87] procedure to control the false discovery rate (FDR) at 0.05 (or, 0.01) for all the methods. In the Results section, $p_{\text{adjust}}$ refers to the adjusted $p$-values. It was shown by Zhu et al. (2021) [38] that parametric methods like SpatialDE and SPARK often produce highly inflated $p$-values for most ST datasets, and hence need additional testing correction. To check if our $p$-values were inflated in the four real datasets, we randomly permuted the spatial locations of the cells/spots five times and then performed the tests using the three methods. Thus, we obtained the empirical null distribution of the $p$-values for each method which we displayed as quantile-quantile plots (see Fig B in S1 Text). In all four cases, SMASH showed no sign of inflation with rather slightly conservative $p$-values which is expected since the minimum $p$-value combination rule used for combining the $p$-values in our method, is known to be conservative [88].

The cosine or periodic kernel covariance matrix is not positive definite (PD). Our testing framework and the distributional derivations hold only for PD kernel covariance matrices. One solution could be to truncate the negative eigenvalues of the kernel matrix, i.e., adjusting $K_S = \sum_{i=1}^{N} \lambda_i U_i U_i^T$ as $K_S^* = \sum_{i=1}^{N} \max\{\lambda_i, 0\} U_i U_i^T$, where $\lambda_i$ and $U_i$ denote the $i$-th eigenvalue and eigenvector, respectively. However, computing eigenvalues can become computationally challenging as it requires a complexity of $O(N^3)$. In our simulations, we have noticed that using unadjusted versions of the kernel matrices yielded conservative test results, with no sign of $p$-value inflation. We refer to S1 Text for further details and plots.

## Enrichment scores

In the real data analysis, we computed the enrichment scores (ES) of the three methods following the procedure outlined in Liu et al. (2022) [48]. Cell clustering based on biological knowledge or using popular software, such as RCTD [32] and Seurat [33], with the transcriptional profiles, can often identify spatially localized layers or cell types. Therefore, marker genes in those spatially-restricted cell types should ideally be identified as SVGs. Suppose there are $M$ cell types. For every cell type $m$, the gene set $G_m$ is built from the top 50 markers based on the fold change between the expression in the cell type $m$ compared to the others. The SVGs detected by the three methods are ranked from the most to the least significant. Finally, unweighted gene set enrichment analysis [89] is implemented to evaluate the enrichment of the gene sets, $G_m$, $m = 1, \ldots, M$, in the high ranking of the ranked SVG lists of the methods.

### Softwares used

To fit SPARK-X, SpaGene, and SpatialDE, we used the existing packages which are available at,

- SPARK-X: https://github.com/xzhoulab/SPARK,

- SpaGene: https://github.com/liuqivandy/SpaGene, and

- SpatialDE: https://github.com/Teichlab/SpatialDE.

Gene-set functional enrichment analyses were performed using ShinyGO Version 0.77 [90] available at, http://bioinformatics.sdstate.edu/go/.

### Simulation description

In simulation setup (1), we generated the spatial coordinates for varying numbers of cells, $N = 1000$, 5000, and 10,000 using a random point-pattern Poisson process [91]. The expression values of $K = 500$ genes in these cells were simulated based on a negative binomial distribution displaying one of the four spatial patterns: streak, reverse streak, hotspot, and reverse hotspot as shown in Fig 1. For each of the patterns, 80% of the spatial locations were assumed to be background locations, while the rest 20% were assumed to be part of the pattern. The difference between the mean expression of a gene on a background location and a patterned location was captured through a fold-change parameter. Several values of fold-change were considered where a value of 1 implied a null scenario i.e., no spatial pattern, and a high value implied a prominent spatial pattern. We refer to Zhu et al. (2021) [38] for more details.

For simulation setup (2), we considered the Gaussian process (GP)-based spatial regression model from SpatialDE [34]. The locations were simulated based on Uniform distribution, which were then used to construct Gaussian covariance matrices with varying lengthscale ($l$) parameters as in Eq (1). The expression levels of genes were independently and identically simulated from the multivariate normal distribution described in Eq (1) for different values of the variance parameters $\tau_k^2$ and $\sigma_k^2$. We fixed the total variance, $\tau_k^2 + \sigma_k^2 = 1$, and varied the individual values as $\tau_k^2 = h$ and $\sigma_k^2 = 1 - h$, where "effect-size" $h$ ranged from zero to larger values implying null to an increasingly stronger spatial pattern. In simulation setup (3), we followed setup (2) replacing the Gaussian covariance with the cosine covariance for varying values of the period parameter $p$. In all three setups, we compared SMASH, SPARK-X, and SpaGene in terms of type 1 error and power.

### Supporting information

**S1 Text.** Section 1 discusses how to choose suitable kernel covariance matrices and combine the $p$-values corresponding to different kernel covariance matrices. Section 2 shows SPARK-X's equivalence with the multiple linear regression model. Section 3 analyzes the null QQ plots of different methods in the real datasets. Section 4 discusses the severity of using non-positive definite (non-PD) kernel covariance matrices. We list and briefly describe the figures from S1 Text below.

- **Fig A.** Visualization of patterns of different kernel covariance matrices.

- **Fig B.** QQ-plots of different methods under null simulations in the real datasets.

- **Fig C.** QQ-plots with the observed and theoretical distributions of the SMASH test statistic with an unadjusted cosine kernel matrix.

- **Fig D.** QQ-plots with the observed and theoretical distributions of the SMASH test statistic with an adjusted cosine kernel matrix.

- **Fig E.** QQ-plots with the observed and theoretical distributions of the—$\log_{10}(p)$-values obtained using SMASH with all the kernel matrices.
  (PDF)

## Acknowledgments

We would like to thank Dr. Kristen Wells-Wrasman for her help with processing the SCCOHT dataset.

## Author Contributions

**Conceptualization:** Souvik Seal, Benjamin G. Bitler, Debashis Ghosh.

**Formal analysis:** Souvik Seal.

**Funding acquisition:** Souvik Seal.

**Investigation:** Souvik Seal, Debashis Ghosh.

**Methodology:** Souvik Seal, Debashis Ghosh.

**Project administration:** Debashis Ghosh.

**Resources:** Benjamin G. Bitler, Debashis Ghosh.

**Software:** Souvik Seal.

**Supervision:** Debashis Ghosh.

**Validation:** Souvik Seal.

**Visualization:** Souvik Seal.

**Writing – original draft:** Souvik Seal, Debashis Ghosh.

**Writing – review & editing:** Souvik Seal, Benjamin G. Bitler, Debashis Ghosh.

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
