## [Decision Letter · Decision Letter 0]

31 May 2023

Dear Dr Seal,

Thank you very much for submitting your Research Article entitled 'SMASH: Scalable Method for Analyzing Spatial Heterogeneity of genes in spatial transcriptomics data' to PLOS Genetics.

The manuscript was fully evaluated at the editorial level and by independent peer reviewers. The reviewers appreciated the attention to an important problem, but raised substantial concerns about the current manuscript. In particular, both reviewers questioned the innovation of the proposed method. What is proposed seems to be highly similar to SPARK-X. Further clarifications on the methodological innovations need to be provided. In addition, real data analyses didn't fully demonstrate why the proposed method is needed. Based on the reviews, we will not be able to accept this version of the manuscript, but we would be willing to review a much-revised version. We cannot, of course, promise publication at that time.

If you decide to revise the manuscript for further consideration at PLOS Genetics, please aim to resubmit within the next 60 days, unless it will take extra time to address the concerns of the reviewers, in which case we would appreciate an expected resubmission date by email to plosgenetics@plos.org.

We are sorry that we cannot be more positive about your manuscript at this stage. Please do not hesitate to contact us if you have any concerns or questions.

Yours sincerely,

Mingyao Li

Academic Editor

PLOS Genetics

Xiaofeng Zhu

Section Editor

PLOS Genetics

Reviewer's Responses to Questions

**Comments to the Authors:**

Reviewer #1: the referee report is uploaded as an attachment

Reviewer #2: This paper presents a non-parametric scalable method, named SMASH, to detect spatially variable genes (SVG) from spatial transcriptomics data. It comes with a Python package available on GitHub. The authors claim that SMASH achieves a good balance between computational burden and statistical power. Please see my major comments below.

1. Even at the beginning of the introduction, the authors made an inappropriate statement. Some widely used sequencing-based spatially resolved transcriptomics (SRT) technologies, such as 10x Visium cannot achieve a much finer spatial resolution than those imaging-based SRT technologies. The authors need to do more research in this field, and know the strengths of different platforms.

2. There are many more SVG detection methods that have been recently developed. However, the authors ignore them. A more comprehensive literature survey is required.

3. Although much details were provided in the “A brief overview of existing methods” section, the authors still used two page on explaining SpatialDE, SPARK, SPARK-X, and SpaGene, which is not necessary. Instead, the authors just need to briefly present their ideas, pros, and cons, especially in terms of time complexity (like Table 1). The derivation of complexity needs to be fully explained or cited.

4. All figures are in super-low quality. They are too blur to get any messages. In Figure 1, it is observed SMASH has a marginal improvement than SPARK-X. The visualization of the simulation results needs to be improved. I don’t think line plot is a good choice under this circumstance.

5. SMASH is a natural extension of SPARK-X, where the distance covariance matrix D in the test statistics, T=tr(E_k D)/N has been replaced by kernel-based distance covariance matrix. Thus, the contribution is very minimal in the field.

6. The authors directly use the results “the asymptotic null distribution of T_k can be well approximated by a gamma distribution” without validating it in the simulation study. Actually, the results were from an unpublished paper on arXiv 10 years ago. I doubt if the approximation is valid in the context of SVG detection, especially when N is large.

Zhang K, Peters J, Janzing D, Sch ¨olkopf B. Kernel-based conditional independence test and application in causal discovery. arXiv preprint arXiv:12023775. 2012;.

7. Lack of details about how to specify the data-driven fixed values of the length-scale characteristic parameter l, which has a huge impact on the kernel-based covariance matrix.

8. The real data analyses were poorly presented, and were not persuasive why the method is needed.

**Have all data underlying the figures and results presented in the manuscript been provided?**

Reviewer #1: Yes

Reviewer #2: **No: **I am not sure. But the figures are all in very poor quality.

PLOS authors have the option to publish the peer review history of their article (what does this mean?). If published, this will include your full peer review and any attached files.

Reviewer #1: No

Reviewer #2: **Yes: **Qiwei Li

---

## [Decision Letter · Decision Letter 1]

19 Sep 2023

Dear Dr Seal,

We are pleased to inform you that your manuscript entitled "SMASH: Scalable Method for Analyzing Spatial Heterogeneity of genes in spatial transcriptomics data" has been editorially accepted for publication in PLOS Genetics. Congratulations!

Yours sincerely,

Mingyao Li

Academic Editor

PLOS Genetics

Xiaofeng Zhu

Section Editor

PLOS Genetics

Comments from the reviewers (if applicable):

Reviewer's Responses to Questions

**Comments to the Authors:**

Reviewer #1: They have addressed my questions.

Reviewer #2: Thank the authors for fully addressing my concerns. I do not have any further comments.

**Have all data underlying the figures and results presented in the manuscript been provided?**

Reviewer #1: Yes

Reviewer #2: None

PLOS authors have the option to publish the peer review history of their article (what does this mean?). If published, this will include your full peer review and any attached files.

Reviewer #1: No

Reviewer #2: No

**Data Deposition**

http://datadryad.org/submit?journalID=pgenetics&manu=PGENETICS-D-23-00398R1

**Press Queries**

---

## [Editor Report · Acceptance letter]

16 Oct 2023

PGENETICS-D-23-00398R1 

SMASH: Scalable Method for Analyzing Spatial Heterogeneity of genes in spatial transcriptomics data 

Dear Dr Seal, 

We are pleased to inform you that your manuscript entitled "SMASH: Scalable Method for Analyzing Spatial Heterogeneity of genes in spatial transcriptomics data" has been formally accepted for publication in PLOS Genetics! Your manuscript is now with our production department and you will be notified of the publication date in due course.

With kind regards,

Judit Kozma

PLOS Genetics

On behalf of:
